# The Survival of Family Farms: Socioemotional Wealth (SEW) and Factors Affecting Intention to Continue the Business

**Manel Plana-Farran** * and **José Luis Gallizo**

Faculty of Law, Economics and Tourism, Universitat de Lleida, 25001 Lleida, Spain; joseluis.gallizo@udl.cat
* Correspondence: manel.plana@udl.cat; Tel.: +34-973-703-206

**Abstract:** This article addresses the problem of succession in family farms in a context of generational change. Family businesses are characterized by their long-term orientation and by having a positive effect through environmental goals that remain in place generation after generation. The general increase in average age among farmers is seen as a barrier to more sustainable land use, and the survival of family farming therefore depends on the availability of a successor in the family. Socioemotional wealth (hereafter, SEW) is understood as the affective endowment of family members. This study adopts the SEW dimensions conceptually validated to analyse the effects of psychological and socioeconomic factors on potential successors' intentions. The results of a survey administered to students attending agricultural schools in Catalonia show that intentions to assume the management and ownership of the family farm increase in line with individuals' interest in creating their own business, their ability to take over the farm, and their emotional inclination to continue the family legacy. In addition, SEW was measured in relation to the potential successor and not the incumbent, as has typically been the case in previous work, bringing this important research subject as a principal actor. Finally, an empirical validation of a short FIBER scale, i.e., REI scale, was obtained that relates individuals' intentions to succeed the family farm to the socioemotional wealth of business families, testing suitability of the REI scale as a measure of intention to succeed.

**Keywords:** family farms; succession; continuity; socioemotional wealth

## 1. Introduction

The intention to continue the business is essential for a smooth transition in management and the success of family businesses [1]. If we refer to succession in family farms (hereafter, FFs), this is conditioned by the possibility of designating a successor or heir [2,3]. According to Suess-Reyes et al. [4], "The future of family farming does not only depend on the farm's adaptability to changing environments, but also on the family's and specially the next generation's sense of attachment to the business and its intention to successfully carry the family's heritage on into the future." As such, the debate and research about intention to succeed FFs among the next generation is the cornerstone for researchers and policy makers. In this vein, one of the main challenges with FFs obtaining transgenerational success is adequately preparing the next generation for providing continuity for the farms in the near future [5].

According to Haberman and Danes [6], FFs are unique businesses in terms of their continuity, values, and expectations for succession, and they have their own characteristics that condition their behaviour in decision-making. Rojas and Lorenzo [7] state that appointing a successor to guarantee the continuity of the farm positively moderates the effect of low profitability, while increasing the effect of setting environmental goals. Family farming accounted for more than 19 of 20 farms across the EU [8], and similar statistics are seen for Catalonia and Spain in particular. The Eurostat Farm Structure Survey [8] shows that 55.8% of European farmers are over 55 years of age, 31.4% are older than 65, and only 6% are younger than 35. This indicates the challenge to the future of rural development,

i.e., the "young farmer problem", which refers to the poor generational renewal rates in the farming sector in the EU [9].

Family businesses are "characterized by visible and active owners, long-term orientation, collective identity, family values, emotional ownership, and a desire for the firm to persist across generations" [10]. It is highly unlikely that the agricultural activity on the farm will be assumed by an outside buyer, given the high investment necessary and the low expected profitability [11], which is why families have the responsibility of designating a successor to ensure the continuity of the farm. Chiswell [12] states that farming is the most hereditary of occupations. The succession process is a concern among rural entrepreneurs and is key to the survival of FFs [13]. In some cases, farmers are undecided about starting the succession process, choosing a successor, or assigning responsibilities. To help the process, different public administrations are developing strategies to support successors in gaining greater awareness of the links between issues related to succession, business longevity, and increased competitiveness [4].

Innovation capacity can be built through education and training, and a special focus on women and youth [14,15] is highly recommended. These strategies are most deeply rooted in agricultural schools, known as Agricultural Training Schools in Spain, and grouped within the "European-International" School Network in Europe (https://europea.org (accessed on 2 April 2021)). Among the aims of agricultural schools is that of encouraging young people to enter the field by training them to guarantee the profitability of their farms, recover self-esteem, and explain to society the importance of food production and the services provided by farms. These educational centres constitute a meeting point for young people who have made the decision to dedicate their working lives to agriculture and who, in many cases, come from farming families. FFs are, in general, intergenerational, so continuity depends on the availability of a successor in the family [16–18].

Education of heirs is another point that influences their decision to continue the FFs. According to Aldando-Ochoa et al. [19] and Hennessy and Rehman [20], formal education of the heirs of FFs decreases the probability of succession. However, as stated by Chiswell [12], research does not consider the intention of successors, or factors affecting the decision regarding FFs.

In view of the above situation, the research question would be: What are the psychological and socioeconomic factors that influence succession intention related to family farms? In Spain and particularly in Catalonia, this research question is important because there is a problem of lack of vocation among young children of farmers, which could cause a crisis of continuity for family farms. It is important for us to know the psychological and socioeconomic factors that influence the transmission of control of family farms to the next generation.

This study attempts to address the intention to continue the FFs in the case of young members of family farms who study in agrarian schools. We used a set of data obtained from 156 students from 13 agricultural schools in Catalonia that pertain to family farming. This article addresses the intention of the interested party to succeed the family business, investigating how SEW impacts on the will of continuing FFs and adopts the SEW literature as a framework for identifying the psychological and socioeconomic factors involved in making the decision to assume the succession [10,21]. In addition, a scale measuring SEW, named REI (a short version of FIBER scale)—known as affective endowment in the context of a family business—was validated for application with next-generation FFs' members.

This research makes a contribution to the research on family farms and the intention to continue them by empirically investigating how some of the main attributes of potential successors (i.e., gender, entrepreneurship, considering agriculture as a viable career future, and affective endowment) influence the intention of these successors to continue the family farm. The results show that the ability and inclination of the potential successor to remain in the family business—with men more likely than women—and to assume responsibility for the management of the business are the most influential factors. The results obtained

are relevant for identifying the succession intention of the heirs and for facilitating the succession process in farming families.

Based on the responses obtained from the survey, a comparative inferential analysis was performed on the intention to continue the family farm according to the intention factors expressed by the respondents. Subsequently, a factor analysis was carried out to establish a relationship between the aforementioned intention and the SEW of the respondents belonging to family farms. The rest of the article is organized as follows: Section 2 provides a review of the previous literature and develops the theoretical framework on which the research is based. The sample of respondents and the applied methodology are explained in Section 3. Section 4 describes the inferential analysis and presents the results obtained from the study, relating the intention to succeed to SEW. Finally, in Section 5 we discuss the main findings and conclusions of our study and propose areas for future research.

## 2. Background and Factors Involved in the Intention to Continue the Family Farm

### 2.1. Family Farms: The Need to Manage a Generational Change

Family farms in Catalonia and around the world are currently experiencing very serious economic, social, and environmental changes [4]. Food safety, sustainability of production, increasing demands on quality standards, and changes in customer tastes have contributed to a new scenario in which the agricultural sector is forced to act, and young farmers must assume a leadership role [22].

More than 90% of the world's farms and almost 75% of agricultural land are managed by families [23]. The reasons range from FFs' aim to satisfy the needs of the family [24] to the use of family labour that allows for the adjustment of labour intensification [25]. Farm tasks are carried out by family members in a more efficient, motivated way and with the ability to understand the local environment [26]. Diversification to agricultural or nonagricultural companies, intensification, or specialization are identified as key strategies for adapting and facing the demands of the market and the pressures of the environment [27–29].

Generational change on farms is driven by required innovations in agricultural processes, new technologies in agricultural work, and climate change, all of which have direct consequences for the production model. The agrifood sector is therefore facing a situation of change, challenge, and the need for constant adaptation that affects all its actors, especially the potential successors of family farming [30]. At the same time, the new generations do not see the agricultural world as attractive for their professional future. Previous studies have pointed out the progressive decrease in the number of young people joining in agricultural activities, an issue that is attributed to economic and social considerations [17,26,31]. This lack of new interest in agriculture has led to a progressive increase in the age of farmers and a lack of generational change in family farms [18]. Therefore, the following subsections discuss the importance, benefits, and potential drivers of young family members' intentions to continue FFs in a changing scenario.

### 2.2. The Challenge of Succession and the Intention to Succeed

The problem of succession in an estate is normally decided by the family, which chooses between heirs; however, the consent of the chosen person is essential for accepting the responsibility for the management of a business. Chiswell [12] has suggested that succession in family farms be analysed from the perspective of potential or future successors, rather than seeing them as marginal figures.

The low proportion of young people engaged in agriculture is a problem that affects rural sustainability and concerns both government institutions and professional agricultural organizations. Zagata and Sutherland [32] have argued that the low proportion of young people on farms is perceived as a loss of potential when it comes to creating more efficient, competitive, and sustainable farms in accordance with the demands derived from the environment and the sector itself. In the EU, the continuous decline in the number of young farmers is considered one of the main drawbacks of EU agriculture [33].

Succession planning is critical to the life of a family farm. It has been observed that, when families cannot identify an heir, they either divest or develop a static management mode, without progress, while farmer families that manage to identify an heir face a variety of horizontal and vertical growth challenges that threaten the continuity of the farm in the future [2].

In our use, intention is the attitude that individuals have towards taking over the family farm. According to different theories, the stronger the intention, the more likely this behaviour is to take place. However, it is well known that the lack of requirements regarding behavioural skills, or the presence of environmental restrictions, can prevent people from acting according to their intentions [34]. "Only when people have control over behavioural performance is intention expected to be a good predictor of behaviour" [34].

In farm succession research, Duesberg et al. [35] conducted a qualitative study to understand the perceptions of farmers without a successor regarding various land transfer options, while Morais et al. [18] investigated to understand the beliefs underlying successors' intention to take over the farm. One possible use of the results of this study would be to help farmer families to identify their heirs early by identifying the determining characteristics of their intention to continue the business.

## 2.3. Factors Influencing the Intention to Continue the Family Farm

While, in many families, there are no explicit rules about succession, the decisions families ultimately make include factors related to the number of successors, their gender, the order of birth, their dedication, and their personal management skills [36].

Based on previous studies [37], we have identified the following factors that influence the intergenerational succession of rural agricultural enterprises.

In a masculinized agricultural society, the gender of the successor plays a prominent role in the succession process. The agricultural sector is a very male-oriented field [38]. Gender is one of the first individual identities; being natural, it cannot be chosen, and it influences behaviour in more categorical ways that can be constructed later [39]. In general, women are not perceived as eligible successors, especially when there are male children. However, in recent years there has been a correction of this trend that needs to be verified [40].

The age of the potential successor is also something to consider. Younger participants who intend to continue their studies may think that their future will not be on the family farm [41].

Commitment, understood as the intention of a potential successor to manage the family farm, was identified as an important factor in previous research. The acquisition of the family business was positively valued, and there were also positive perceptions regarding the business management capacity and social pressure to control the family property [18]. In our work, we asked respondents about their particular interest in continuing the family business, understanding that this is one of the main elements in the decision-making process—that is, people are determined to dedicate their lives to the activities of agricultural and livestock management.

The formal and informal knowledge of the agricultural environment and of the particular farm can increase the intention of succession of the heir. It has been proven that specific knowledge of the farm generates an incentive for young people to assume the responsibility of continuing with the agricultural business [42,43], and that children's interest in agriculture is a factor to consider when analysing the characteristics of potential and/or future successors of family farms [44]. Likewise, previous research has revealed the need for complementarity of formal and informal knowledge in the agricultural sector. Specifically, although local experience is valued, a new formal knowledge base is also required, with new content and learning processes necessary for the transition to a more sustainable agriculture [45]. Young people trained in this new knowledge will be more willing to assume the generational transfer [45].

### 2.4. The Relationship between Intention of Potential Successors and Socioemotional Wealth (SEW)

Research has found that "family businesses limit the goal of maximizing profits in exchange for maintaining control of the business and passing that control on to future generations" [46]. In addition to economic ownership, FFs have affective values that influence the succession process; these family business values have been grouped under the concept of socioemotional wealth. SEW is understood as the "affective endowment of family owners" [47]; that is, the benefits unrelated to profit that a family obtains from its position as the owner of a company [47–49]. Unlike nonfamily farms, family-owned farms attend to both economic and affective values in their decision-making process, and this is the case of property transfer, where family entrepreneurs have the incentive to preserve control for future generations [50].

This article looks at the SEW of potential successors to family farms. We assume that the desire to preserve SEW motivates family farm owners to retain their legal ownership and that this SEW goes beyond its financial objectives. This is the case with those family farm owners who show a commitment to preserving the tradition of agriculture in the region. For them, following a conservation strategy acts as a substitute to ensure the emotional attachment of their successors to agriculture [51].

One aim of this study was to contribute to the literature on SEW and family businesses by considering that an owner's SEW should be studied in the cultural context in which the family business exists. The young farmers who have participated in this study were trained through formal and regulated programs at agricultural schools and learned values of attachment to the land and respect for the environment, with the aim of joining the family farming business and taking charge of it. These students' intention to take over the business is indicated by their willingness to continue the family tradition and the model of life represented by working on the farm. These features corroborate the existence of an "affective patrimony" that is not only in the hands of the family owners but also in the hands of the potential and future successors. This leads us to suppose the existence of a relationship between the heir's intention to succeed the family business and the emotional wealth of the family [12].

### 2.5. Development of Hypotheses

Identifying the impact of psychological factors and affective values that affect the succession intention of young farmers is key for farming. The nature of SEW permits the capture of the affective endowments and how they affect the intention to continue the FFs. This was our starting point in identifying the impact of psychological and family farm factors on the succession intention of the potential heirs of agricultural holdings. Following it, our approach tried to predict the behaviour of the individual involved in the succession debate, as determined by prospective-successors' views about continuing the family legacy, their sense of belonging to the business family, and their personal commitment, according to SEW, as perceived by the successor—in the assumption of the farm, the individual prioritizes the preservation of property and the protection of the agricultural community over economic interests [51].

In this research, we explored the succession intention via the impact of some psychological factors that are present during the intergenerational succession process in family farming. It is known that a FF's continuity exerts a powerful influence on the development of the farm (Chiswell, 2014). At that moment, the intention to take over the farm will be greater if the successors consider that the following circumstances exist: viability of their professional future, ability to take charge of the farm, and the desire to continue the family legacy (socioemotional wealth). Based on this reasoning, we proposed the following hypotheses:

**Hypothesis 1.** *The affective relationship and belonging to the FFs (SEW) exerts a positive influence on the intention of the successors.*

In order to obtain more information on the factors that influence the intention of agricultural successors in Catalonia, we examined two background control variables: gender and age (Table 1). These two factors were identified in previous studies as drivers influencing the option to succeed the FF and discouraging the emigration of young people away from rural areas [18].

**Table 1.** Comparative inferential analysis. Intention to continue the FFs depending on the gender and age of the participant (*n* = 156).

| FACTOR/Categories | Total Sample (*n* = 156) | Intention to Succeed (%) | | Chi-Squared Test | | Effect Size $R^2$ |
|---|---|---|---|---|---|---|
| | | YES (88.5%) | NO (11.5%) | Value | *p*-Value | |
| Gender | | | | 5.57 * | 0.018 | 0.036 |
| Male | (*n* = 148) | 89.9% | 10.1% | | | |
| Female | (*n* = 8) | 62.5% | 37.5% | | | |
| Age | | | | 1.19 NS | 0.754 | 0.008 |
| 15–20 years | (*n* = 129) | 87.6% | 12.4% | | | |
| 21–25 years | (*n* = 14) | 92.9% | 7.1% | | | |
| 26–30 years | (*n* = 6) | 100% | 0.0% | | | |
| >01 years | (*n* = 7) | 85.7% | 14.3% | | | |

NS = Nonsignificant. * = Significant.

To establish the relationship between the student members of FFs' succession intention and the SEW, an abridged version REI was used [52]. We intend to evaluate the theoretical construct of emotional wealth via the empirical FIBER scale [53].

The FIBER scale in the REI version consists of 16 items in Likert format with five options, configured on three dimensions: (a) R—Renewal of family ties through succession, four items; (b) E—Emotional Relationship of family members, six items; and (c) I—Identification of family members with the company, another six items. Moreover, the REI scale has been revalidated for the population of students of agrarian schools that belong to FFs. A factor analysis by principal components was applied to verify the unidimensionality of each set of items with the dimension to which they theoretically belong.

For the present statistical analysis, and with the intention that a higher score in the variable is interpreted as a higher degree (greater domain) in the dimension evaluated, the responses of the respondents were recoded on a scale of 1 = completely disagree to 5 = strongly agree.

## 3. Methodology

### 3.1. Participants and Context

Our field of action in this research comprised agricultural schools, and we obtained the data of young people planning to join the farming business, who in most cases belonged to farming families.

The target group was 161 students belonging to family farms who enrolled in Catalan agricultural schools. The dataset was based on agricultural school students who were training to work in agriculture and belonged to family farms. Thirteen agricultural schools are present in the region, and they depend on the Departments of Agriculture and Education. The training courses are recognized by the Catalan Department of Agriculture and give qualified students eligibility for EU young farmers' subsidies. Of the total of 161, 5 students had to be excluded from the study because they did not answer some of the questions. Thus, the final sample consisted of 156 students. Participants' age ranged from 15 to 31 years old. The most common age was 15 to 20 (71.9%).

### 3.2. Procedure

In order to reach as many students as possible, a schedule was agreed with the management of each agrarian school. We visited each school (thirteen in total, around all Catalonia), explaining the objective of the study. Participation was anonymous and not mandatory. In order to avoid any potential misunderstandings or mistakes, we were present during the process of survey completion. The data collection was completed during 2019.

### 3.3. Instruments and Measures

In the present investigation, a questionnaire with different sections was used. A section about demographic variables was utilized in order to ask the students about their gender, age, and the intention to succeed FFs. An additional section was used to assess the student's situation with regard to his or her intrinsic factors. Finally, the last section was the REI scale, which measured three dimensions (i.e., Renewal of family ties through succession; Emotional attachment of family members; Identification of family members with the company).

### 3.4. Statistical Analysis

The obtained data were analysed using Statistical Product and Service Solutions (SPSS) version 22 (IBM Corporation, 2017, IBM SPSS Statistics v 25.0 for Windows; Armonk, NY, USA). The distribution of frequencies and percentages was used with the qualitative (nominal) variables, with an estimate of 95% confidence intervals. For quantitative variables, the data were explored using the Q–Q plot for normality fit, histogram, coefficients of skewness, and kurtosis/height, together with the Kolmogorov–Smirnov goodness-of-fit test and description with the usual tools of centrality (mean, median) and variability (standard deviation and range). Regarding the reliability of our measurement scale, this was estimated with Cronbach's α coefficient and intraclass correlation.

Exploratory factor analysis by principal components was used to discover the internal structure of variables being used to detect the relationship between the different variables. Moreover, the Mann–Whitney–Wilcoxon test, a nonparametric test that contrasts whether two samples proceed from an evenly distributed population, was used to test significant differences between the means of groups, for example, when comparing intention to succeed FFs with level of SEW. In addition, the effect size was estimated using $R^2$ and the chi-square test, for the crossover of categorical variables ($R^2$ estimated from Cramér's V). Finally, the usual 5% level of significance was established (significant if $p < 0.05$), except in the KS goodness-of-fit test, where only serious deviations were considered significant, that is, at 1% ($p < 0.01$).

## 4. Results

### 4.1. Inferential Analysis

In the first place, we proceed to describe this sample of students attending agricultural schools, all of whom were related to a family farm. The majority of students in the general sample who came from FFs were men: 94.9%. The majority were between 15 and 20 years old: 82.7%. Additionally, the majority (88.5%) were conducting their studies to continue with the farm of the family, based on question that sought to capture the intention to succeed.

The gender of the potential heir was one of the first factors to consider. There was a statistically significant relationship ($p < 0.05$) between the intention to continue with the FFs and sex (a slight effect of 3.6%), such that we can admit that this intention was higher among men than among women (89.9% vs. 62.5%). However, there was no statistically significant relationship ($p > 0.05$) with age (almost null effect: <1%). Therefore, there was no evidence that the succession intention was linked to the student's age, this intention always being above 85.7% (Table 1).

*4.2. Revalidation of REI scale in the Context of Family Farming*

The REI scale was revalidated for the population of Agrarian Schools' students who belong to FFs. The methodology of factor analysis by principal components was used to verify the unidimensionality of each set of items within the dimension to which they theoretically belonged. The degree of reliability of each of these dimensions was also estimated using the classic Cronbach's α coefficient, as well as of each of the items with the item-total homogeneity index. The results obtained are summarized in the tables below.

Three dimensions were obtained that grouped a set of items that measured the basic affective gifts that a family can derive from the control of a company. The degree of reliability of each of these dimensions was also estimated, using Cronbach's α coefficient, as well as of each of the items with the item-total homogeneity index.

1. Dimension R (Figure 1) refers to the renewal of family ties through succession, where the items include continuing the family legacy and tradition, valuing the family's investment as long-term, and the transfer of the family farm to the next generation being an important goal for family members.

| Item | Statement | Scale (1-5) |
|---|---|---|
| SEW R1 | Continuing the family legacy and traditional is an important goal for my family business. | Strongly desagree –totally agree |
| SEW R2 | Family owners are less likely to evaluate their investment on a short-term basis | Strongly desagree –totally agree |
| SEW R3 | Family members would be unlikely to consider selling the family business | Strongly desagree –totally agree |
| SEW R4 | Successful business transfer to the next generation is an important goal for family members | Strongly desagree –totally agree |

**Figure 1.** Statements and scales used for the measurable items that represent SEW dimension R. Source: [54].

The mean values of these four items were high (between 3.82 and 4.43 out of 5), which indicates that the position of the students in our sample clearly tended towards agreement with each proposal made to evaluate this dimension. The values of the communalities were optimal in all the items and the factor loadings were high (between 0.617 and 0.813). These data demonstrate that the unidimensionality of these items with respect to dimension R. Additionally, the reliability indices of the items were sufficient (>0.300), so that the reliability of this dimension was good (0.664), (Table 2).

**Table 2.** Unidimensionality and reliability. Dimension: R—Renewal of family ties through succession of the REI ($n$ = 156).

| ITEM | Descriptive | Factor Analysis by C.P. | | Reliability |
|---|---|---|---|---|
| | Average (D.E.) | Commonality | Factor Load | |
| SEW—R1 | 4.43 (0.72) | 0.381 | 0.617 | 0.359 |
| SEW—R2 | 3.82 (0.94) | 0.442 | 0.665 | 0.409 |
| SEW—R3 | 4.05 (1.17) | 0.536 | 0.732 | 0.474 |
| SEW—R4 | 4.17 (0.93) | 0.661 | 0.813 | 0.574 |
| Total | 16.47 (2.69) | — | 500.49% | 0.664 |

2. Dimension E (Figure 2) refers to the emotional attachment of family members, considering whether these bonds are strong or not. This dimension reports on the emotional proximity of each of the members with reference to the family and the company.

| Item | Statement | Scale (1-5) |
|---|---|---|
| SEW E1 | Emotions and sentiments often affect decision-making processes in my family business | Strongly desagree –totally agree |
| SEW E2 | Protecting the welfare of family members is critical to us, apart from personal contributions to the business | Strongly desagree –totally agree |
| SEW E3 | . In my family business, the emotional bonds between family members are very strong | Strongly desagree –totally agree |
| SEW E4 | . In my family business, affective considerations are often as important as economic considerations | Strongly desagree –totally agree |
| SEW E5 | Strong emotional ties among family members help us maintain a positive self-concept | Strongly desagree –totally agree |
| SEW E6 | In my family business, family members feel warmth for each other | Strongly desagree –totally agree |

**Figure 2.** Statements and scales used for the measurable items that represent SEW dimension E. Source: [54].

In our results, (Table 3) the mean values of all the items of dimension E were high (between 4.11 and 4.33), indicating favourable opinions of potential successors towards the content of each one. The values of the communalities were very high, and the factorial weights were as well (between 0.730 and 0.825). This demonstrates the one-dimensional belonging of these six items to dimension E of emotional attachment. Likewise, the reliability indices of these items were high (>0.600), which generated a high reliability coefficient (0.857).

**Table 3.** Unidimensionality and reliability. Dimension E—Emotional Attachment of family members of the REI (*n* = 156).

| ITEM | Descriptive | Factor Analysis by C.P. | | Reliability |
|---|---|---|---|---|
| | Average (D.E.) | Commonality | Factor Load | |
| SEW—E1 | 4.11 (0.97) | 0.652 | 0.807 | 0.699 |
| SEW—E2 | 4.33 (0.76) | 0.539 | 0.734 | 0.611 |
| SEW—E3 | 4.17 (0.82) | 0.680 | 0.825 | 0.722 |
| SEW—E4 | 4.13 (0.83) | 0.533 | 0.730 | 0.606 |
| SEW—E5 | 4.16 (0.84) | 0.539 | 0.734 | 0.614 |
| SEW—E6 | 4.31 (0.86) | 0.567 | 0.753 | 0.629 |
| Total | 25.22 (3.89) | — | 58.49% | 0.857 |

3. Dimension I (Figure 3) refers to the identification of family members with the agricultural exploitation, where the members have a strong sense of belonging and a great personal commitment towards the business, to the point of feeling proud to communicate to others that they belong to the family business.

| Item | Statement | Scale (1-5) |
|---|---|---|
| SEW I1 | Family members have a strong sense of belonging to my family business | Strongly desagree –totally agree |
| SEW I2 | Family members feel that the family business' success is their own success. | Strongly desagree –totally agree |
| SEW I3 | My family business has a great deal of personal meaning for family members | Strongly desagree –totally agree |
| SEW I4 | Being a member of the family business helps define who we are | Strongly desagree –totally agree |
| SEW I5 | Family members are proud to tell others that we are part of the family business | Strongly desagree –totally agree |
| SEW I6 | Costumers often associate the family name with the family business's products and services | Strongly desagree –totally agree |

**Figure 3.** Statements and scales used for the measurable items that represent SEW dimension I. Source: [54].

The mean values of the six items of the dimension that summarize the identification of the family members were high (between 3.95 and 4.33), which implies a high degree of agreement with the statements. The communalities were high (>0.500), and the factor

loadings were also high (between 0.670 and 0.804). Consequently, we can conclude the unidimensionality of this set of items with respect to dimension I. Likewise, the reliability indices of these items were high (>0.500), so the reliability coefficient of the full dimension was also high (0.810) (Table 4).

**Table 4.** Unidimensionality and reliability. Dimension I—Identification of family members with the company of the REI (*n* = 156).

| ITEM | Descriptive | Factor Analysis by C.P. | | Reliability |
|---|---|---|---|---|
| | Average (D.E.) | Commonality | Factor Load | |
| SEW—I1 | 3.95 (1.07) | 0.531 | 0.729 | 0.596 |
| SEW—I2 | 4.12 (0.92) | 0.449 | 0.670 | 0.538 |
| SEW—I3 | 4.32 (0.84) | 0.546 | 0.739 | 0.580 |
| SEW—I4 | 4.19 (0.91) | 0.647 | 0.804 | 0.666 |
| SEW—I5 | 4.33 (0.82) | 0.492 | 0.701 | 0.536 |
| SEW—I6 | 4.01 (1.02) | 0.447 | 0.668 | 0.526 |
| Total | 24.90 (4.02) | — | 51.86% | 0.810 |

In summary, the obtained results adequately guarantee (1) the belonging of these items to the theoretically expected dimension, (2) the sufficient reliability of each of the items, and (3) the high reliability of each of the items.

*4.3. Confirmatory Factor Analysis for Applicability of REI Scale*

In order to revalidate the use of the REI scale in the population, a confirmatory factor analysis was used. Figure 4 presents the validated model, with the three dimensions intercorrelated with each other, together with the items that we have just demonstrated to belong to each of them.

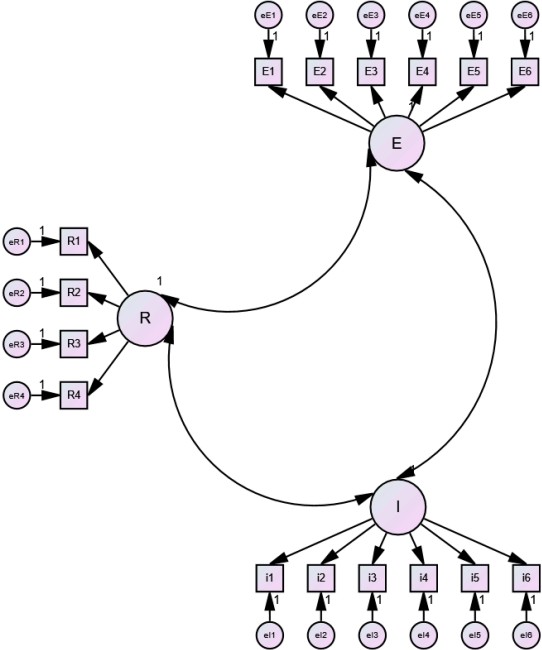

**Figure 4.** Correlation diagram of the confirmatory factor analysis. Items and dimensions are from the REI scale. *n* = 156 students from agrarian schools with FFs.

The fit of the data with the model was studied using the goodness-of-fit index (Table 5). The value of the RMSEA index was found to be below the 0.080 cutoff, with both its lower value (0.065) and almost all of its CI (95%): 0.047–0.082; therefore, it does not generate doubts about the optimal fit of the data to the model. In the same sense, the other indices

(NFI, IFI, TLI, CFI) exceeded the minimum cut-off (0.800) and generally even the value 0.900 was considered as a large adjustment. Finally, the chi-square test ratio was less than 3, which confirmed the good fit. In short, the fit of the theoretical model that we were trying to test with the empirical data was good enough so that the result of the CFA obtained could be considered as reliable.

**Table 5.** *CFA: Goodness-of-fit*. Questionnaire REI (*n* = 156 students of Agrarian Schools, members of FFs).

| Model | RMSEA | NFI | IFI | TLI | CFI | Cmin/DF |
|---|---|---|---|---|---|---|
| *Three* dimensions | 0.065 | 0.850 | 0.935 | 0.921 | 0.934 | 1.65 |

This CFA result is summarized in Table 6 for clarity of discussion. We can verify that: (1) the items have high standardized coefficients (>0.500) in the dimension to which they must belong, and (2) and the dimensions are highly related to each other as proposed in the model.

**Table 6.** *Confirmatory Factor Analysis.* REI Scale (*n* = 156 students of Agrarian Schools, members of FFs).

| N° Ítem | PART | AU.LID | AP.SIG |
|---|---|---|---|
| SEW—R 1 | 0.518 | | |
| SEW—R 2 | 0.596 | | |
| SEW—R 3 | 0.510 | | |
| SEW—R 4 | 0.710 | | |
| SEW—E 1 | | 0.788 | |
| SEW—E 2 | | 0.648 | |
| SEW—E 3 | | 0.764 | |
| SEW—E 4 | | 0.684 | |
| SEW—E 5 | | 0.679 | |
| SEW—E 6 | | 0.682 | |
| SEW—I 1 | | | 0.630 |
| SEW—I 2 | | | 0.586 |
| SEW—I 3 | | | 0.697 |
| SEW—I 4 | | | 0.763 |
| SEW—I 5 | | | 0.664 |
| SEW—I 6 | | | 0.551 |
| **DIMENSIONES** | **R** | **E** | **I** |
| **R** | – | 0.948 | 0.871 |
| **E** | 0.948 | – | 0.855 |
| **I** | 0.871 | 0.855 | – |

*4.4. REI Dimensions and Intention to Continue the FFs*

In the next stage, we studied the relationship between the variables of these three REI dimensions and succession intention, for which a contrast test of the significance of the difference in means was carried out. Due to the lack of adjustment to normality, a nonparametric method was chosen, the Mann–Whitney U Test. The existence of significance implies that there was a relationship between SEW and the succession intention and that the REI variables were explanatory factors of it. The contrast test was also accompanied by the estimation of the effect size in $R^2$, from the value of Cohen's *d*. The results are summarized in Table 7.

**Table 7.** Comparative inferential analysis. Differences in the SEW based on the students' succession intention (*n* = 156).

| DIMENSION VARIABLES | Total Sample (*n* = 156) | Student Succession Intention (%) | | Mann–Whitney Test | | Effect Size $R^2$ |
|---|---|---|---|---|---|---|
| | | Yes (88.5%) | Not (11.5%) | | *p*-Value | |
| R—Renewal of family ties through succession | | | | 3.25 ** | 0.001 | 0.101 |
| Mean (standard deviation) | 16.47 (2.69) | 16.78 (2.44) | 14.11 (3.39) | | | |
| Minimum/Max. values | 7/20 | 8/20 | 7/20 | | | |
| E—Emotional relationships of family members | | | | 4.38 ** | 0.000 | 0.189 |
| Mean (standard deviation) | 25.22 (3.89) | 25.83 (3.16) | 20.56 (5.58) | | | |
| Minimum/Max. values | 8/30 | 11/30 | 8/28 | | | |
| I—Identification of family members with the company | | | | 3.66 ** | 0.000 | 0.139 |
| Mean (standard deviation) | 24.90 (4.02) | 25.44 (3.52) | 20.78 (5.17) | | | |
| Minimum/Max. values | 9/30 | 15/30 | 9/29 | | | |

** = Highly significant.

Dimension R. It was observed that the mean value in this variable was higher in the group of subjects who intended to continue with the FFs (16.8 vs. 14.1), a difference that we must consider as highly significant with *p* < 0.01 and that corresponds to a moderate–high effect size (10.1%).

Dimension E. Once again, it was found that the mean value of the participants who answered that they would continue with their EAF was higher than those who said they would not (25.8 vs. 20.6). This difference is highly significant (*p* < 0.001) and is equivalent to an effect size that is already considered large (18.9%).

Dimension I. Similar to the previous dimensions, in this final dimension, it was again found that the average values of the respondents who intended to take over the family business was higher than the average of those who said they would not continue (25.4 vs. 20.8). This difference was also highly significant (*p* < 0.001) and equivalent to an effect size that, although somewhat smaller than the previous one, is still large (13.9%).

To contrast the H1, given the positive influence of the SEW on the intention of the successors, we have developed an abridged version of the REI [54], empirically revalidated according to the results obtained in the preceding section. With this version, we intend to evaluate the theoretical construct of emotional wealth with the empirical validation of the scale. (Figure 5)

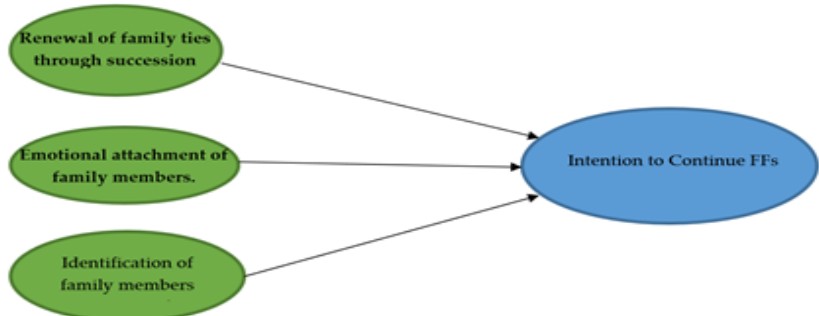

**Figure 5.** REI scale and intention to continue FFs. A comparative inferential analysis. Intention to continue the FFs depending on SEW student's level.

## 5. Discussion and Conclusions

This study analysed the intention to take over the family business among students who belonged to a family farm and attended agricultural school. It was based on the theoretical framework of the SEW literature and the application of REI dimensions to

analyse the intention to succeed FFs [54]. The results obtained from the comparative inferential analysis on intrinsic factors related to the student indicated that gender strongly influences these students' intention to take over the family farm. Moreover, the rest of the results obtained shed light on a key aspect for renewal and continuity in farming, i.e., the commitment of these students to succeed FFs.

This study revealed some significant findings that may contribute to the literature on both family farming and family business. More specifically, this study found that the SEW scale (REI) fits well for studying family-farming future-successors; moreover, the results have shown that level of SEW affects positively the intention to continue family farms. According to these results, a relationship was observed between the SEW level of students and the intention of succession. Specifically, we obtained: (a) a high degree of agreement with the belief in the renewal of family ties through dynastic succession, (b) a high degree of emotional attachment of family members to FFs, and (c) a high degree of family identification with the company. The high factorial scores explain that the existence of a sense of belonging and commitment of the young FF members positively influence the succession intention of the potential successors. Hence, we state these factors are facilitators of succession intention.

The REI scale allowed us to use an instrument designed to analyse the level of socioemotional wealth (SEW) to test H1 with respect to the intention to take over the family farm. From this analysis, three factors were obtained that summarize the influence of the affective endowments of the family on the control and transfer of the company. The first, "family ties", is related to the importance of continuing the family legacy and tradition. In this regard, it was found that prioritizing succession and family elements over economic motivations played a major role in determining the intention of the possible successors to succeed the family farm. Staying in control and transferring the business to future generations is an important motivation among FFs' holders to extend the useful life of the farm. For Inwood and Sharp [2], agricultural families that prioritize succession and noneconomic domestic factors implement a wide range of growth and innovation strategies in agricultural production that, ultimately, contribute to the persistence and resilience of the farm. In this dimension, we observe that prioritizing succession and family elements over economic motivations are characteristics that give a high weight to the succession intention of potential successors.

The second factor, "emotional attachment", obtained a high score for all its items, showing that the existence of an affective relationship between family members makes it more likely that a potential successor will decide to accept the inheritance. The emotional bond of the family has been considered important in the succession process; in fact, in previous research it has been shown that trust and communication between family members have a strong influence on the succession process [54–57]. According to the study by Morris et al. (1997) [58], problems within the family and emotional detachment were the main factors that led to 60% of failed successions among the companies studied. The third factor, "identification of relatives with the FF", explains the high influence that the sense of belonging and the personal commitment of the members of a family-run agricultural business have on the intention to continue the farm.

In the end, the results obtained determine the intrinsic and socioemotional factors of agricultural school students that facilitate the process of incorporation to FFs. Knowledge of these factors is useful to better understand the generational transition process. The practical implications of this study include the actions that parents and older relatives can take with respect to future generations. These should be actions aimed at helping potential successors to obtain greater knowledge of the business, encouraging them to share the objectives of the company, and transmitting a sense of belonging, thereby enhancing the affective relationship of family members. These are elements that must be present in farming families to improve the intention of succession of their heirs.

Our insights into the intention to continue FFs and bring the successor into focus offer relevant and insightful managerial implications. Increasing attention to economic, social,

psychological, and emotional aspects of FFs' successors is capital for the present and future of family farming. This idea goes in line with the findings shown by Arzubiaga et al. [59], who specifically highlighted the importance of taking care of the socio-emotional aspects of new successors when facing and designing succession processes.

This study has some limitations that offer opportunities for future research. This study focuses on a region, Catalonia, and on farming school students. Therefore, any conclusions should be interpreted carefully in other regions [60], in other FFs' potential successors, or in other family farming collectives. As such, these results call for different settings to theorize about emotional attachment and continuity in family farming, and future research should take advantage of integrating findings across different contexts and regions, deepening and building more cumulative research results. FFs' succession is supported by considerable research in domains other than the domain of the potential successor, [12]; however, more research is needed to enrich the family farming literature from an emotional and psychological perspective. For example, future research is needed to assess whether and to what extent our findings apply to other agrarian regions or to family farming principals. Our findings indicate the opportunity of future research to explore the emotional implication and commitment to family farming. The level of FFs' continuity represents a challenge for the future of farming.

**Author Contributions:** All the authors contributed to conceptualization, formal analysis, investigation, methodology, and writing and editing the original draft. All authors have read and agreed to the published version of the manuscript.

**Funding:** This research received support from the Family Business Chair of the University de Lleida.

**Institutional Review Board Statement:** Informed consent was obtained from the respondents of the survey.

**Informed Consent Statement:** Informed consent was obtained from all subjects involved in the study.

**Data Availability Statement:** The data will be made available on request from the corresponding author.

**Acknowledgments:** We want to thank the support received from the Catalan Ministry of Agriculture, Departament d'Agricultura, Ramaderia, Pesca i Alimentació.

**Conflicts of Interest:** The authors declare no conflict of interest.

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
