# Peer review of "The Survival of Family Farms: Socioemotional Wealth (SEW) and Factors Affecting Intention to Continue the Business"

_agriculture, doi:10.3390/agriculture11060520_

Round 1
Reviewer 1 Report
The authors clearly state the aim of the article. As well, the methodology they follow looks scientifically well-founded. The empirical evidence and the consideration developed in this article can be used for decisions by agricultural policy makers. Based on these observations, I positively evaluate this article.
However, the support for young farmers is important within the EU rural development policies because it concerns a critical issue in EU agriculture – an aging farm population. This policy measure is also important because it absorbs a significant share of financial resources. Therefore, the authors could offer recommendations for more effective policy measures in the field of young farmers.
Author Response
Dear reviewer,
Thank you for your interest and the insightful comments on the earlier version of our manuscript. We apreciate all your time in reviewing our study.

Reviewer 2 Report
Overall Comments:
The main purpose of this study was to determine, within the reasoned action approach, which factors influence intergenerational succession on family farms. The research questions are unclear throughout the manuscript. The Literature Review did not present why the authors focus on the topics and variables of this study, and how these variables were examined in the previous studies on family farm succession. One of the strength was the use of RAA as a framework for identifying the psychological and socioeconomic factors and farm characteristics involved in making the decision to assume succession. However, authors did not clearly explained how the RAA has informed the study. The measures of the variables for the manuscript should be explained in sufficient detail and the sampling strategy is not clear. The results section is confusing since the research questions and statistical methods were not clearly presented in previous sections. My detailed comments are below.
1. Introduction:
The explanation of family farm succession was helpful. However, the motivation would be stronger if it was more clearly tied to the Spain family farms. The first three paragraphs bring the important issue of why this study examines this topic; but the rest of this section is mainly the summary of the study that highlights the sample, conceptual framework, methods, and findings. What are the main research questions of this study? Why are these research questions important in Spain? In the Introduction, the authors could address the current situation of succession planning in Spain and what has been or is currently being done about this topic. The rationale and scientific objectives of the study (e.g., research questions) should be clearly stated at the end of the Introduction.
2. Background and factors in the intention to succeed family farms:
Overall, the literature review was incomplete under this section. The authors mentioned specific factors that influence a family farm successor, such as number of successors, gender, birth order, dedication, and personal management skills. However, each of these different factors should be reviewed based on the literature.
Influencing factors in the intention to succeed the family farm - In this section, only three factors (e.g., gender, commitment, and informal and formal knowledge) were addressed. However, the authors mention multiple factors, such as number of successor, that are later included in their analyses. There should be more background on each of these variable for the reader to understand why they are being examined. Also, if commitment is the same as dedication, this section should be “Dedication”. If the “Informal and formal knowledge” section is the same as personal management and agricultural skills, this heading should be changed to reflect this. These sections should be labeled consistently with variables examined.
The relationship between the intention to succeed and socio-emotional wealth: Since the authors did not clearly present key research questions of this study, it was difficult to understand this section. The second two paragraphs of this section did not seem to fit under this heading. This section could provide information on previous research in the factors (e.g., psychological, socioeconomic factors, and the characteristics of the farm) that influence successors’ intentions to succeed the business. The authors need to include a more comprehensive literature review before the Methods section.
One of my major concerns with this paper is how the reasoned actions approach that the authors mentioned in their introduction and literature review sections is explained in this paper. It could be helpful to create a dedicated “Conceptual Framework” section that helps the readers understand how this approach is applied to the current study and how it guides this study to examine factors affecting intention to succeed. A set of hypotheses can be presented at the end of this section.
3. Sample and Methodology:
The beginning of this section was confusing and difficult to read. There was not information pertaining to the data, data collection, the sampling procedures, the time frame of the data, and so on. There should be a dedicated “Data” section and a separate “Sample” section, instead of eight small paragraphs. It was very distracting to read the section this way.
This section should specifically explain the variables the authors used in this study, such as age, gender, number of successors, socio-emotional wealth, etc. However, the authors did not explain which variables they measured and how, which could help the readers to follow the results of the study.
A concise account of all statistical methods must be given in this section. In particular, it was difficult to understand the results presented under the heading – the Relationship between Intention to Succeed FF and the SEW.
Results of the comparative inferential analysis – I am wondering why the results are presented under the methodology section as they reported the results presented in the Tables 1 and 2. These Tables are also not well designed and not clearly presented. The authors did not provide much in terms of descriptive results. Table 1 only includes age and gender; it did not include any other individual, family, or business characteristics. There was no explanation as to why the variables in Table 2 were used. The authors should address these variables in the literature review and explain this reasoning of inclusion in the methods section to help the readers understand why Table 2 is necessary and important.
The heading of section 3.1.2 is also unclear. The sample seemed to be agricultural students, but this section examines factors related to the family farm owner. If the authors feel these results are important, they should introduce previous literature and information on the FF owner in the literature review section.
4. Relationship between Intention to Succeed FF and the SEW:
Results must be presented in logical order, and tables could be better designed. Throughout the manuscript, it seems that the authors are exploring how SEW can predict the intention to succeed FF. There is no explanation of the research design to understand if SEW is the independent or dependent variable. Clear research questions, hypotheses, and variable explanation in previous sections could make this more easy to understand.
Also, using the conceptual framework of RAA, it is unclear how the relationship between intention to succeed FF and SEW is examined. Furthermore, the authors suddenly introduce the REI scale, which was not mentioned in the methods section. This scale should be explained in the methods section to help readers understand what it measures. In the tables, the authors only write “SEW-R1” and so on, which does not clearly explain what each question is measuring.
5. Discussion and Conclusions:
In this section, the results must be discussed point by point, in logical order and concisely in the light of the topic literature. That is, the findings of the study must be interpreted and discussed in a wider context with appropriate literature citations. What are the findings from this study? Are there any new or conflicting findings? If so, they should be highlighted. The authors restated some results, but these results should be expanded on and interpreted as to why these were the results and what other factors could be related to these results.
Author Response
Dear reviewer,
We would like to thank your interest and time in reviewing our study. We are also grateful for your insightful and useful comment on the earlier version of our manuscript.
Thank you again

Reviewer 3 Report
The topic is interesting but it absolutely confusing.I am not really sure what is it about (see my comments below). If editors decide to revise this manuscript my remarks have to be taken into account.
- Introduction
- state clear aim at the end of Introduction
Lines 62-63: „which comprises 54% of the target population“ – target population for this research are all possible successors for family farms in Spain, net your students
Lines 59-74: this is not information for Introduction - delete it
- Background . . .
- add hypotheses to be tested
- add graphical presentation of your model
Line 82-82: citation is needed
Line 138-139: citation is needed
Lines 140-149: please cite literature to the age of new farmers
- Sample . . .
- description of questionnaire is missing – please add information which questions were asked and add them to the overall model from „2. Background . . .“, readers do not know how intention, attitude, perceived norms and perceived behavioural control were measured.
Lines 203-210: add how the respondents were selected or all 700 students were addressed and 374 agreed to participate?
Table 1: I am really confused – n=156, how it is possible – there were 374 respondents
Table 2: only here we see what the respondents were asked and how answers were measured – this must be in questionnaire description
Parts 3.1 and 3.2 – this should be in „Results“ Citations are also here – this discussion with former results should be in Discussion part.
Chi-square test for ordinal variables is not appropriate here – perhaps Mann-Whitney test is better for your data.
- Relationship between Intention . . .
Line 289-300: So, here is the model used – please move this to methods and in “2. Background” write way this model was used and what it its importance for theory of reasoned action. The ties between intention and three predictors (attitude, perceived norms, behavioural control) are solved in model of theory of reasoned action. Here REI is used. It is not clear why.
Tables 4, 6, 7 (table 5 is missing or the numbering is wrong): what “Carga Factorial” is? – seems like “factor loadings”, how can you calculate reliability for single items? From my point of view REI should be analysed as the whole not using factor analysis based on PCA for those three “dimensions” separately.
. . . AND where is the analysis of REI and “intention to succeed the business”???????? The main analysis is missing!!!!!!!!!!!!
- Discussion and conclusions
This part is not discussion, it is summary of results.
Author Response

(The authors gave the same response as above.)

Round 2
Reviewer 2 Report
Dear authors,
The revision of the manuscript “The Survival of Family Farms: Factors Affecting Intention to Continue the Business” is good. Overall, great improvements have been made to this manuscript. You have removed or added information as needed and the main idea of this study is more clear. In the Introduction, you emphasized the current issue of farm succession in Spain. The changes made to the Literature Review are helpful and the organization is more clear than the first submission. The new data section better explains the study data and analyses.
There are still a couple issues with this manuscript that could be addressed. First, while you rearranged the Results section and edited important information, you still have personal interpretations and discussion under this heading. On the other hand, the actual Discussion section is still relatively short. I would suggest moving any interpretation and discussion of your hypotheses and results to the Discussion section. This is my main concern. Another small issue I had was that it was not clear what EF meant when mentioned at the beginning of the paper. Please make sure that any terms you use that may be unfamiliar to the reader are clearly defined.
Author Response
Dear reviewer
We really feel really grateful for all your support and insightful comments that have allowed us developing a more robust and compelling paper. We hope that we have taken full advantage of your feedback and that by so doing, the paper has reached a higher level. Thank you!
As you can state, we have updated the title of the work, adapting to final version.
We really appreciate all your insights comments.

Reviewer 3 Report
please see the attached file

Author Response
Dear reviewer,
Thank you for recognizing the potential in our work and for providing us a new opportunity to improve the manuscript. Following your advice, we have adapted this new version to your comments. I believe that is a fruitful process and allows us to improve the article.
